# Antioxidant Mechanisms of Echinatin and Licochalcone A

**DOI:** 10.3390/molecules24010003

**Published:** 2018-12-20

**Authors:** Minshi Liang, Xican Li, Xiaojian Ouyang, Hong Xie, Dongfeng Chen

**Affiliations:** 1School of Chinese Herbal Medicine, Guangzhou University of Chinese Medicine, Guangzhou 510006, China; lminshi@outlook.com (M.L.); oyxiaojian55@163.com (X.O.); xiehongxh1@163.com (H.X.); 2Innovative Research & Development Laboratory of TCM, Guangzhou University of Chinese Medicine, Guangzhou 510006, China; 3School of Basic Medical Science, Guangzhou University of Chinese Medicine, Guangzhou 510006, China; 4The Research Center of Basic Integrative Medicine, Guangzhou University of Chinese Medicine, Guangzhou 510006, China

**Keywords:** antioxidant, echinatin, licochalcone A, 1,1-dimethyl-2-propenyl, α,α-dimethyl-β-propenyl, radical adduct formation, dimer

## Abstract

Echinatin and its 1,1-dimethyl-2-propenyl derivative licochalcone A are two chalcones found in the Chinese herbal medicine *Gancao*. First, their antioxidant mechanisms were investigated using four sets of colorimetric measurements in this study. Three sets were performed in aqueous solution, namely Cu^2+^-reduction, Fe^3+^-reduction, and 2-phenyl-4,4,5,5-tetramethylimidazoline-1-oxyl 3-oxide radical (PTIO•)-scavenging measurements, while 1,1-diphenyl-2-picrylhydrazyl radical (DPPH•)-scavenging colorimetric measurements were conducted in methanol solution. The four sets of measurements showed that the radical-scavenging (or metal-reduction) percentages for both echinatin and licochalcone A increased dose-dependently. However, echinatin always gave higher IC_50_ values than licochalcone A. Further, each product of the reactions of the chalcones with DPPH• was determined using electrospray ionization quadrupole time-of-flight tandem mass spectrometry (UPLC-ESI-Q-TOF-MS/MS). The UPLC-ESI-Q-TOF-MS/MS determination for echinatin yielded several echinatin–DPPH adduct peaks (*m*/*z* 662, 226, and 196) and dimeric echinatin peaks (*m*/*z* 538, 417, and 297). Similarly, that for licochalcone A yielded licochalcone A-DPPH adduct peaks (*m*/*z* 730, 226, and 196) and dimeric licochalcone A peaks (*m*/*z* 674 and 553). Finally, the above experimental data were analyzed using mass spectrometry data analysis techniques, resonance theory, and ionization constant calculations. It was concluded that, (i) in aqueous solution, both echinatin and licochalcone A may undergo an electron transfer (ET) and a proton transfer (PT) to cause the antioxidant action. In addition, (ii) in alcoholic solution, hydrogen atom transfer (HAT) antioxidant mechanisms may also occur for both. HAT may preferably occur at the 4-OH, rather than the 4′-OH. Accordingly, the oxygen at the 4-position participates in radical adduct formation (RAF). Lastly, (iii) the 1,1-dimethyl-2-propenyl substituent improves the antioxidant action in both aqueous and alcoholic solutions.

## 1. Introduction

Echinatin and licochalcone A (Figure 1) are two chalcones in the Chinese herbal medicine *Gancao* (*Glycyrrhizae radix* et rhizoma) [1,2]. A recent study suggested that both chalcones can be absorbed by human Caco-2 monolayer cells after the oral administration of *Gancao* [3]. However, pure echinatin and licochalcone A have been shown to exhibit hepatoprotective [4] and anti-inflammatory effects [5,6]. These pharmacological effects are known to be closely associated with antioxidant action [7,8]. A literature survey revealed that no relevant studies have focused on the antioxidant effects (or mechanisms) of echinatin or licochalcone A; thus, it is necessary to conduct such a study. In this study, the antioxidant potentials of the 4-OH and 4′-OH groups in each chalcone molecule were compared for the first time.

As shown in Figure 1, echinatin and licochalcone A both bear a 4′-OH group in the *A* ring, and a 4-OH group in the B ring. These two phenolic -OH groups possess similar but different chemical environments (Figure 1). Similar situations can also be found in other chalcones and isoflavones, such as licochalcone C (**2a**, Figure 2), corylifol A (**2b**, Figure 2), 2′-*O*-methylisoliquiritigenin, oureirin A, daidzein, daidzin, 7,4′-di-*O*-methyldaidzein, 8-prenyldaidzein, 3′-methoxypuerarin, 4′-methoxypuerarin, and isoformononetin (Appendix A). However, whether there is a difference in the antioxidant abilities of the two phenolic -OHs remains unknown [9,10,11,12]. Therefore, the present study will also deepen our understanding of these chalcones and isoflavones.

As illustrated Figure 1, licochalcone A is an echinatin derivative with a 1,1-dimethyl-2-propenyl substituent (also called α,α-dimethyl-β-propenyl). If there is an antioxidant activity difference between echinatin and licochalcone A, then it can be attributed to the existence of the 1,1-dimethyl-2-propenyl substituent. As such, parallel studies of echinatin and licochalcone A can well determine whether and how the 1,1-dimethyl-2-propenyl groups affect antioxidant chalcones, and even phytophenols. In fact, 1,1-dimethyl-2-propenyl substituents are widely distributed in phenolics (Appendix A), including xanthones (e.g., isocudraniazanthone A (**2c**, Figure 2) [13]), naphthoquinones (e.g., 6,8-dihydroxy-2,7-dimethoxy-3-(1,1-dimethylprop-2-enyl)-1,4-naphthoquinone (**2d**, Figure 2) [14]), isoflavones (e.g., 5,7,4′-trihydroxy-8-(1,1-dimethylprop-2-enyl) isoflavone (**2e**, Figure 2) [15]), and isoflavanes (e.g., glycyuralin B (**2f**, Figure 2) [1]). Undoubtedly, the present study will also help us to determine whether the 1,1-dimethyl-2-propenyl substituent affects the antioxidant potentials of these phenolics.

For these purposes, a set of measurements was designed as experiments for determining the antioxidant activity in this study. These antioxidant activity measurements were cupric-ion (Cu^2+^)-reduction (pH 7.4), ferric-ion (Fe^3+^)-reduction (pH 3.6), 2-phenyl-4,4,5,5-tetramethylimidazoline-1-oxyl 3-oxide radical (PTIO•)-scavenging (pH 7.4), and 1,1-diphenyl-2-picrylhydrazyl radical (DPPH•)-scavenging colorimetric measurements. The reaction products from the DPPH•-scavenging measurements were further investigated using ultra-performance liquid chromatography coupled with electrospray ionization quadrupole time-of-flight tandem mass spectrometry (UPLC-ESI-Q-TOF-MS/MS). Because the mechanisms of these antioxidant activity measurements are distinctive and complementary, it is believed that this study will bring reliable experimental results. 

## 2. Results and Discussion

The antioxidant mechanisms of phenolics are suggested to involve electron transfer (ET) [16], proton transfer (PT) [17], proton-coupled electron transfer (PCET) [18], concerted proton electron transfer (CPET) [19], ET-PT (or sequential electron proton transfer, SEPT) [20], PT-ET (or sequential proton loss electron transfer, SPLET) [21], or hydrogen atom transfer (HAT) pathways [22,23,24,25]. These antioxidant mechanisms are seemingly different from each other; however, all of these are essentially made up of two elemental reactions, i.e., an ET reaction and a PT reaction. The difference between them lies only in the sequence and collaboration. 

To test the possibility of ET, echinatin and licochalcone A were individually examined using a Fe^3+^-reduction measurement at pH 3.6. To determine whether a pH of 3.6 suppressed PT (i.e., phenolic-OH ionization), we calculated the molar concentration of deprotonated echinatin (*c*, Figure 3 and Equation (1)) [13].
(1)$$\mathrm{Ka}=\frac{\left[H^{+}\right]\cdot c}{\left(\left[\mathrm{Echinatin}\right]-c\right)}$$

As shown in Figure 3, echinatin has two potential H^+^ ionization sites, i.e., the 4′-OH and 4-OH groups. If 4′-OH ionization occurs, echinatin-4′-*O*-monoanion, one form of deprotonated echinatin, is produced (Figure 3A). However, the pKa values for the 4′-OH and 4-OH groups of echinatin are not available; thus, we consulted the pKa value of its analogue, hydroxyacetophenone, which is 7.87 [26]. At pH 3.6, the ionization equilibrium constant equation was as follows (Equation (2)):(2)$$10^{-7.87}=\frac{\left[10^{-3.6}\right]\cdot c}{\left(\left[\mathrm{Echinatin}\right]-c\right)}$$

In our experiments, the tested concentrations of echinatin varied from 70.1 to 370.4 μM (Appendix A). When the concentration of echinatin (i.e., [echinatin]) was 70.1 and 370.4 μM, the echinatin-4′-*O*-monoanion concentrations (*c*) were calculated to be 3.76 × 10^−6^ and 1.99 × 10^−5^ μM, respectively, using Equation (2). In these cases, the deprotonation percentages were 0.0053% (3.76 × 10^−6^/70.1 or 1.99 × 10^−5^/370.4). 

However, if 4-OH ionization occurs, echinatin-4-*O*-monoanion, another form of deprotonated echinatin, is produced (Figure 4A). It is roughly equivalent to curcumin, which has been indicated to have a pKa of 8.38 [27]. Correspondingly, the ionization equilibrium constant equation can be expressed as Equation (3). When [echinatin] was 70.1 and 370.4 μM, the echinatin-4-*O*-monoanion concentrations (*c*) were calculated to be 1.16 × 10^−6^ and 6.15 × 10^−6^ μM, respectively, using Equation (3). Hence, its deprotonation percentages were 0.0016% (1.16 × 10^−6^/70.1 or 6.15 × 10^−6^/370.4).
(3)$$10^{-8.38}=\frac{\left[10^{-3.6}\right]\cdot c}{\left(\left[\mathrm{Echinatin}\right]-c\right)}$$

The values of 0.0053% and 0.0016% suggested that both the 4′-OH and 4-OH groups in echinatin, respectively, are not deprotonated in a pH 3.6 buffer. Regardless, the two pKa values (7.86 and 8.38) were somewhat inaccurate, and relevant calculations for licochalcone A have not been performed; however, the conclusion that a pH of 3.6 effectively suppressed H^+^ ionization (i.e., deprotonation and PT) is generally reasonable. This is also in agreement with the previous literature [28]. On the other hand, our experimental data showed that echinatin and licochalcone A efficiently reduced Fe^3+^ to Fe^2+^ at pH 3.6 (Appendix A and Table 1). This means that both echinatin and licochalcone A have ET potentials under conditions where H^+^ ionization is suppressed. The ET potentials of echinatin and licochalcone A can explain why chalcones can alter the redox status of Jurkat T cells [29].

Similarly, the deprotonation percentages of echinatin at the physiological pH of 7.4 can also be calculated by using the relevant formula. The calculation revealed that, at pH 7.4, the 4′-OH and 4-OH groups in echinatin showed 25.3% and 9.5% deprotonation, respectively. This means that PT for echinatin and licochalcone A is facilitated at pH 7.4. For further investigation, echinatin and licochalcone A were evaluated using the Cu^2+^-reduction method at pH 7.4. The results showed that the Cu^2+^-reduction percentages for both echinatin and licochalcone A increased in a dose-dependent manner (Appendix A). In fact, the Cu^2+^-reduction measurements have been reported to involve ET and PT. However, the destination of the PT is the solution rather than the Cu^+^ complex [30,31]. 

In order to assess whether they can transfer H^+^ to a free radical, both echinatin and licochalcone A were examined using PITO•-scavenging colorimetric measurement, a method that was newly established by our team [17]. During PITO• scavenging, an electron and a proton were transferred to PITO• to form PITO-H [32]. The observations that echinatin and licochalcone A effectively scavenged PITO• radicals (Appendix A) implied that they both have ET and PT potential in aqueous solution. This can partially explain the fact that aqueous extracts of *Gancao* as an antioxidant show a protective effects in a renal hypoxia (ischemia) reoxygenation (reperfusion) model [33].

The reason that both echinatin and licochalcone A exhibited PT potential may be that the Cu^2+^-reduction and PITO•-scavenging experiments were conducted in aqueous solutions. Water, which is strongly polar (dielectric constant ε = 78.36 F/m), can force phenolic -OH groups to ionize (deprotonate) via solvation. In contrast, organic molecules are weakly polar (e.g., methanol ε = 32.7 F/m) and can force phenolic -OH groups to ionize slightly via solvation. Thus, in methanol solution, phenols may transfer hydrogen mainly through a hydrogen atom (H•) pathway rather than a hydrogen ion (H^+^) pathway. This is known as an HAT mechanism [34]. The fact that a hydrogen atom carries an electron makes HAT a redox-based antioxidant mechanism. Typically, DPPH•-scavenging determination in alcoholic solutions can be used to examine the HAT of a phenolic antioxidant [35,36,37]. 

Now, given that a HAT reaction takes place for echinatin (and licochalcone A), there are only two reaction sites, i.e., 4-OH and 4′-OH. The HAT reaction of the 4-OH group produces an echinatin-4-O• radical, which has six resonance structures (I–VI) (Figure 5); meanwhile, the HAT reaction of the 4′-OH group produces an echinatin-4′-O• radical, which has only five resonance structures (VII–XI). Thus, the echinatin-4-O• radical is more stable, which implies that the 4-OH group has more HAT potential than the 4′-OH group. After an HAT reaction at the 4-OH group, six intermediate radicals (i.e., I–VI) are generated. These radicals may combine with each other via covalent bonding to form radical adduct formation (RAF) products. However, RAF products can be monitored easily using UPLC-ESI-Q-TOF-MS/MS analysis [38]. The main results are shown in Figure 6. 

As shown in Figure 6D, from the product of the reaction between echinatin and the DPPH• radical, at least three chromatographic peaks (1–3) were found through UPLC-ESI-Q-TOF-MS/MS analysis. Subsequent examination of the MS spectra suggested that three chromatographic peaks could each present an *m*/*z* value of 662 (Figure 6D,E). This value, however, was exactly two units less than the sum of the molecular weights of echinatin (M.W. 270) and the DPPH• radical (M.W. 394). Thus, we initially assumed that the four peaks corresponded to echinatin–DPPH adducts. Out of the three peaks, peak 3 could be broken down further to produce an MS/MS spectrum that showed four main fragments (*m*/*z* 662, 615, 226, and 196, Figure 6F). In light of previous evidence [39], the echinatin–DPPH adduct was assumed to be **7a** (Figure 7), and its MS spectrum has been fully elucidated and is shown in Figure 8A. Despite these fragments (Figure 6F), other possible echinatin–DPPH adduct structures can be deduced; however, it is definitive that echinatin and DPPH^•^ radical generated RAF products. 

In addition to echinatin–DPPH adducts, the echinatin–echinatin dimer may also be formed during the reaction of echinatin with the DPPH• radical. The 4-O of the above echinatin intermediate radical (I, Figure 5) links to the 5-C of structure IV, producing an echinatin–echinatin dimer [24], and it is re-numbered as 4″-O (**7b**, Figure 7). Then, **7b** undergoes keto–enol tautomerization to produce the stable echinatin–echinatin dimer **7c** (Figure 7). The structure of **7c** can be further confirmed through full MS spectrum elucidation (Figure 8B). 

The reaction of echinatin with the DPPH• radical yields two types of RAF products (i.e., the echinatin–DPPH adduct and the echinatin–echinatin dimer). The discovery of the two types of RAF products suggested that echinatin scavenges DPPH• radicals through HAT. This is because RAF depends on the covalent bonding between two (intermediate) radicals [40]; however, the production of intermediate radicals depends on the HAT reaction. 

Similarly, licochalcone A also generated three peaks for licochalcone A–DPPH adducts (Figure 9D) and two peaks for the licochalcone A–licochalcone A dimer (Figure 9H) after treatment with the DPPH• radical. One possible structure of the licochalcone A–DPPH adducts is **10a**, while one possible structure of the licochalcone A–licochalcone A dimers is **10b** (Figure 10). The structures of both **10a** and **10b** were fully elucidated through their MS spectra, which are shown in Figure 11. 

Now, it is obvious that, in aqueous solution, the antioxidant mechanisms of echinatin and licochalcone A involve ET and PT; however, in organic solutions, HAT (a combined form of ET and PT) may occur preferentially at the 4-OH group and not at the 4′-OH group. Through HAT antioxidant mechanisms, both chalcones can produce two types of RAF products (i.e., the chalcone–chalcone dimer and the chalcone–radical adduct). 

It should be noted that (1) a higher HAT potential does not mean more PT. As discussed above, it is easier for the 4′-OH group to give H^+^ and to initiate a PT reaction, as it has a lower pK_a_ value or lower ionization potentials (IPs). However, a mere PT reaction cannot transfer electrons. Thus, mere PT potential is relevant to the acidic rather than the antioxidant potential. Furthermore, (2) as shown in Figure 5A, echinatin may link with the 4-O atom at other carbon atoms (e.g., 1-C, 3-C, and α-C), in addition to the 5-C position. These different linkages may result in similar MS spectra, as shown in Figure 7. Similarly, (3) with licochalcone A, other carbon atoms besides α-C (e.g., 1-C and 3-C) can link to the 4-O (Figure 10). However, regardless of the multiple possibilities mentioned in points 2 and 3, it is certainly clear that dimeric echinatin (or licochalcone A) is generated in the DPPH•-scavenging reaction.

Finally, the relative antioxidant activities of echinatin and licochalcone A were quantitatively compared, based on their IC_50_ values. As shown in Table 1, echinatin always gave higher IC_50_ values than those of licochalcone A. Such a difference can certainly be attributed to the 1,1-dimethyl-2-propenyl substituent at the 5-position of licochalcone A. Accordingly, the 1,1-dimethyl-2-propenyl substituent was assumed to enhance antioxidant chalcones. This enhancement is possibly derived from its electron-donating inductive effect (+I). Of course, more work is required for identification in the future. 

## 3. Materials and Methods

### 3.1. Chemicals 

Echinatin (C_16_H_14_O_4_, CAS number 34221-41-5, M.W. 270.2, purity 99%, Appendix A) was obtained from Chengdu Alfa Biotechnology Co., Ltd. (Chengdu, China). Licochalcone A (C_21_H_22_O_4_, CAS number 58749-22-7, M.W. 338.4, purity 98%, Appendix A) was obtained from BioBioPha Co., Ltd. (Kunming, China). Pyrogallol, 2,4,6-tripyridyl triazine (TPTZ), 2,9-dimethyl-1,10-phenanthroline (neocuproine), and (±)-6-hydroxyl-2,5,7,8-tetramethylchromane-2-carboxylic acid (Trolox) were obtained from Sigma-Aldrich (Shanghai, China). 1,1-Diphenyl-2-picrylhydrazyl radical (DPPH•, C_18_H_12_N_5_O_6_) was obtained from Aladdin Chemical, Ltd. (Shanghai, China). The 2-phenyl-4,4,5,5-tetramethylimidazoline-1-oxyl 3-oxide radical (PTIO•) was obtained from TCI Chemical Co. (Shanghai, China). Water and acetonitrile were high performance liquid chromatography (HPLC) grade. FeCl_3_·6H_2_O and other reagents were analytical grade and purchased from Guangdong Guanghua Chemical Plants Co., Ltd. (Shantou, China). 

### 3.2. Fe^3+^-Reduction Antioxidant Colorimetric Measurement 

The Fe^3+^-reduction antioxidant colorimetric measurement used in this study was adapted from the method reported by Benzie and Strain [28]. This measurement can be used to give an indication of the reducing ability of a material or mixture. The measurement was performed in a buffer with a pH of 3.6. Briefly, in a ratio of 1:1:10, the determining reagent was freshly prepared by mixing together 10 mM TPTZ and 20 mM FeCl_3_ in 0.25 M acetic acid–sodium aqueous buffer solution (pH 3.6). The test sample (*x* = 0–10 μL, 0.5 mg/mL) was added to (20 − *x*) μL of methanol, followed by 80 μL of the determining reagent. The absorbance at 593 nm was recorded after 3 h of incubation at 40 °C against a blank consisting of an acetate buffer. The relative reducing power of the sample to the maximum absorbance was calculated using the following formula:(4)$$\mathrm{Relative}\;\mathrm{reducing}\;\mathrm{power}\;\%=\frac{A-A_{\min}}{A_{\max}-A_{\min}}\times 100\%$$
where A_max_ is the maximum absorbance, A_min_ is the minimum absorbance, and A is the absorbance of the sample.

### 3.3. Cu^2+^-Reduction Antioxidant Spectrophotometric Measurement 

The Cu^2+^-reduction measurement was performed based on the method proposed by Apak et al. [30], with the small modifications proposed by Tian [41]. In a 96-well plate, 12 μL of an aqueous CuSO_4_ solution (0.01 M) and 12 μL of a methanol neocuproine solution (7.5 × 10^−3^ M) were added, and they were mixed with samples of different concentrations (0–50 μg/mL). The total volume was then adjusted to 100 μL with an ammonium acetate buffer solution (0.1 M), and the solutions were mixed again to homogenize them. Each mixture was incubated at room temperature for 30 min, and the absorbance was measured at 450 nm on a microplate reader (Multiskan FC, Thermo Scientific, Shanghai, China). The relative reducing power of the sample was calculated using the following formula:(5)$$\mathrm{Relative}\;\mathrm{reducing}\;\mathrm{power}\;\%=\frac{A-A_{\min}}{A_{\max}-A_{\min}}\times 100\%$$
where A is the absorbance of the sample at 450 nm, A_max_ is the maximum absorbance at 450 nm, and A_min_ is the minimum absorbance at 450 nm.

### 3.4. PTIO•-Scavenging Colorimetric Measurement 

The PTIO•-scavenging colorimetric measurement was conducted in accordance with our method [17]. In brief, the test sample solution (*x* = 0–10 μL, 1 mg/mL) was added to (20 − x) μL of methanol, followed by 80 μL of an aqueous PTIO• solution. The mixture was kept at 40 °C for 2 h, and then the absorbance was measured at 560 nm using a microplate reader (Multiskan FC, Thermo Scientific, Shanghai, China). The PTIO• inhibition percentage was calculated as follows:(6)$$\mathrm{Inhibition}\;\%=\frac{A_{0}-A}{A_{0}}\times 100\%$$
where A_0_ is the absorbance of the control without the sample, and A is the absorbance of the reaction mixture with the sample.

### 3.5. DPPH•-Scavenging Colorimetric Measurement 

The DPPH• radical scavenging activity was determined as follows [42]. 80 μL of DPPH• solution (0.1 mM) was added to a 96-well plate and mixed with samples of different concentrations (0–50 μg/mL). The total volume was then adjusted to 100 μL with methanol. The mixture was kept in the dark at room temperature for 30 min, and then the absorbance was measured at 519 nm on a microplate reader (Multiskan FC, Thermo Scientific, Shanghai, China). The DPPH• inhibition percentages of the samples were calculated by using the formula in Section 3.4. 

### 3.6. UPLC-ESI-Q-TOF-MS/MS Analysis of DPPH• Reaction Products with Echinatin and Licochalcone A

In the echinatin experiment, the reaction product was prepared using the method from our previous study [20]. In brief, a methanol solution of echinatin was mixed with a methanol–DPPH• solution at a molar ratio of 1:2, and the resulting mixture was incubated for 10 h at room temperature. The product was then filtered through a 0.22 μm filter for UPLC-ESI-Q-TOF-MS/MS analysis.

The UPLC-ESI-Q-TOF-MS/MS analysis was based on the method described in our previous report [43]. The analysis system was equipped with a C_18_ column (2.0 mm i.d. × 100 mm, 2.2 μm, Shimadzu Co., Kyoto, Japan). A mobile phase was used for the elution of the system, and consisted of a mixture of acetonitrile (phase A) and 0.1% formic acid water (phase B). The column was eluted at a flow rate of 0.2 mL/min with the following gradient elution program: 0–2 min, maintained at 30% B; 2–10 min, 30–0% B; 10–12 min, 0–30% B. The sample injection volume was 1 μL for the separation of the different components. Q-TOF-MS/MS analysis was performed on a TripleTOF 5600^+^ mass spectrometer (AB SCIEX, Framingham, MA, USA) equipped with an ESI source, which was operated in the negative ionization mode. The scan range was 100–2000 Da. The system was operated with the following parameters: ion spray voltage, −4500 V; ion source heater temperature, 550 °C; curtain gas (CUR, N_2_) pressure, 30 psi; nebulizing gas (GS1, air) pressure, 50 psi; and Tis gas (GS2, air) pressure, 50 psi. The declustering potential (DP) was −100 V, while the collision energy (CE) was −40 V with a collision energy spread (CES) of 20 V. The final RAF products were quantified by extracting the corresponding formula (i.e., [C_33_H_21_N_5_O_11_-H]^−^ for the echinatin–DPPH adduct, and [C_30_H_18_O_10_-H]^−^ for the echinatin–echinatin dimer) from the total ion chromatogram and the corresponding peak. 

In the licochalcone A experiment, the above protocols were repeated using licochalcone A. Correspondingly, the extracted formulas were [C_27_H_16_N_5_O_11_-H]^−^ for the licochalcone A-DPPH adduct and [C_18_H_10_O_10_-H]^−^ for the licochalcone A–licochalcone A dimer. 

### 3.7. Statistical Analysis

The colorimetric measurement tests were performed in triplicate. The measurement results are shown as the mean ± SD in the dose-response curves. The IC_50_ values were calculated using linear regression analysis based on the dose-response curves, and independent-sample *T* tests were performed to compare the different groups. A *p* value of less than 0.05 was considered to be statistically significant. Statistical analyses were performed using the software SPSS for Windows version 17.0 (SPSS Inc., Chicago, IL, USA). All of the linear regression analyses described in this paper were processed using the Origin Professional software (2017 version, OriginLab, Northampton, MA, USA). 

## 4. Conclusions

In conclusion, in aqueous solution, both echinatin and licochalcone A may undergo an electron transfer (ET) and a proton transfer (PT) to cause antioxidant action. However, in an alcoholic solution, both of them may prefer HAT to show antioxidant activity. The HAT active site is the 4-OH group rather than the 4′-OH group. Accordingly, both can generate final RAF products, i.e., a chalcone–chalcone dimer and a chalcone–radical adduct. In addition, the 1,1-dimethyl-2-propenyl substituent (α,α-dimethyl-β-propenyl substituent) plays a role in improving the antioxidation reaction in both aqueous and alcoholic solutions.

## Figures and Tables

**Figure 1 molecules-24-00003-f001:**
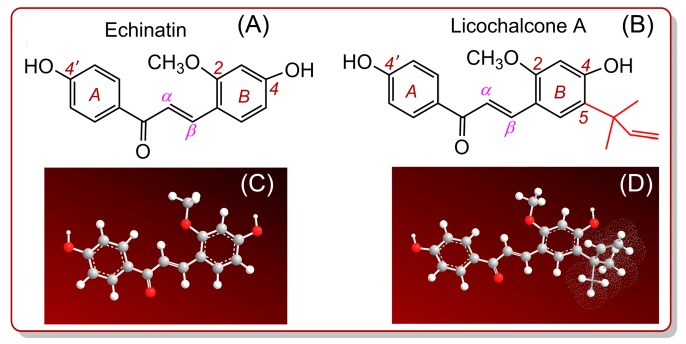
Structural formulas and molecular models of echinatin and licochalcone A: **A**, structural formula of echinatin; **B**, structural formula of licochalcone A; **C**, preferential conformation-based molecular model of echinatin; and **D**, preferential conformation-based molecular model of licochalcone A. The molecular models were created based on molecular mechanics calculations using Chem3D Pro 14.0. (red, O; gray, C; white, H).

**Figure 2 molecules-24-00003-f002:**
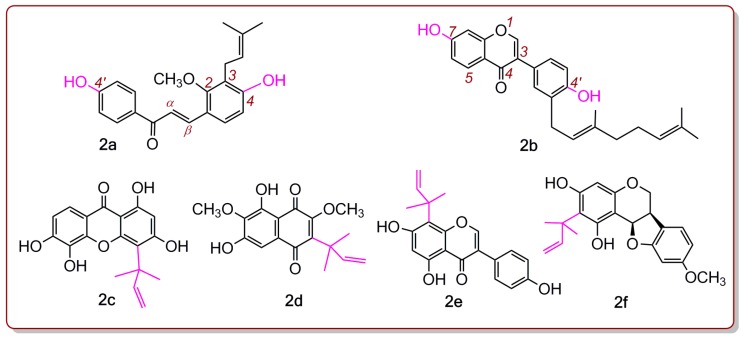
Structures of licochalcone C (**2a**), corylifol A (**2b**), isocudraniazanthone A (**2c**), 6,8-dihydroxy-2,7-dimethoxy-3-(1,1-dimethylprop-2-enyl)-1,4-naphthoquinone (**2d**), 5,7,4′-trihydroxy-8-(1,1-dimethylprop-2-enyl) isoflavone (**2e**), and glycyuralin B (**2f**).

**Figure 3 molecules-24-00003-f003:**
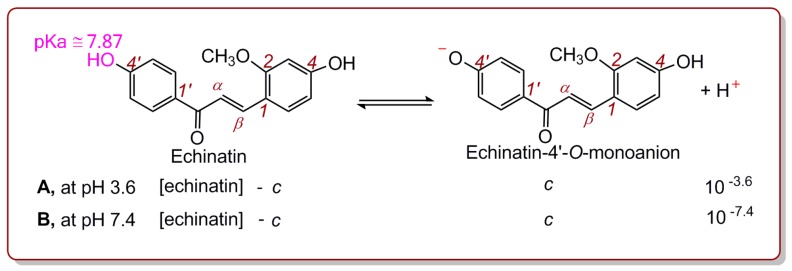
Ionization equilibrium of the 4′-OH group in echinatin at pH 3.6 (A) and 7.4 (B).

**Figure 4 molecules-24-00003-f004:**
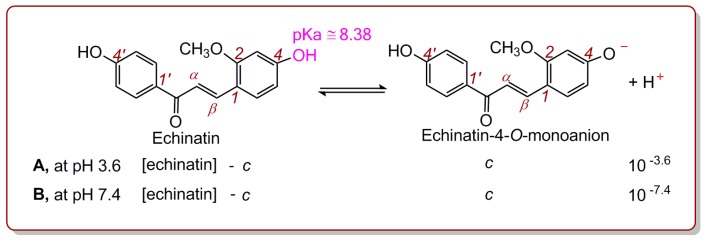
Ionization equilibrium of the 4-OH group in echinatin at pH 3.6 (A) and 7.4 (B).

**Figure 5 molecules-24-00003-f005:**
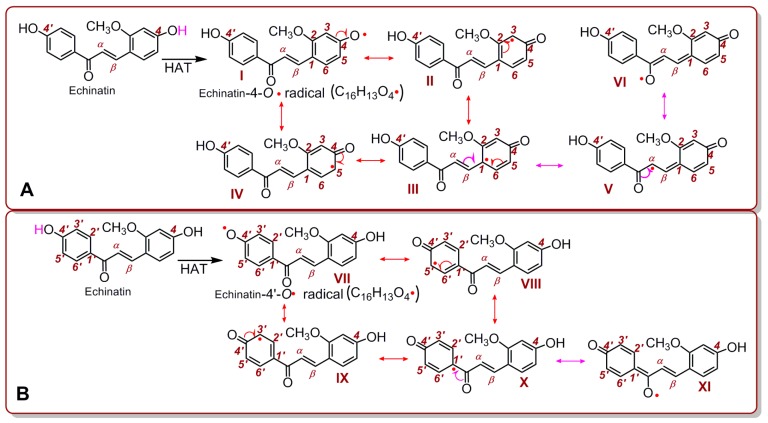
Resonance structures of the echinatin-4-O• radical (**A**) and the echinatin-4′-O• radical (**B**).

**Figure 6 molecules-24-00003-f006:**
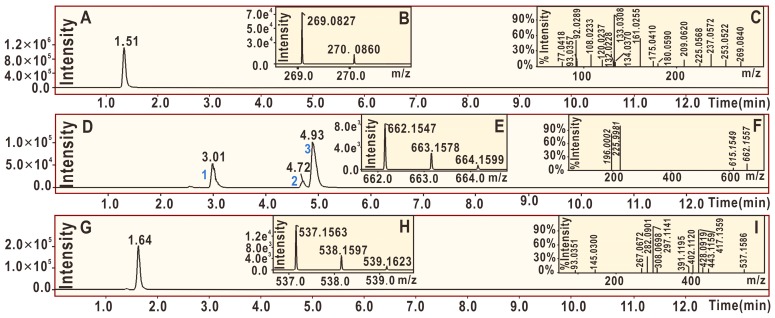
Main results of UPLC-ESI-Q-TOF-MS/MS analysis of echinatin: (**A**) total ion chromatographic diagram of echinatin; (**B**) primary MS spectrum of echinatin; (**C**) MS/MS spectrum of echinatin; (**D**) total ion chromatographic diagram of the RAF products of echinatin and DPPH• radical extracted by [C_34_H_25_N_5_O_10_-H]^−^; (**E**) primary MS spectra of echinatin–DPPH adducts (from peaks 1 to 3 in (**D**)); (**F**) MS/MS spectra of echinatin–DPPH adducts from peak 3 in (**D**); (**G**) total ion chromatographic diagram of the possible dimeric products of echinatin extracted by [C_32_H_26_O_8_-H]^−^; (**H**) primary MS spectrum of the echinatin–echinatin dimer; and (**I**) MS/MS spectrum of the echinatin–echinatin dimer.

**Figure 7 molecules-24-00003-f007:**
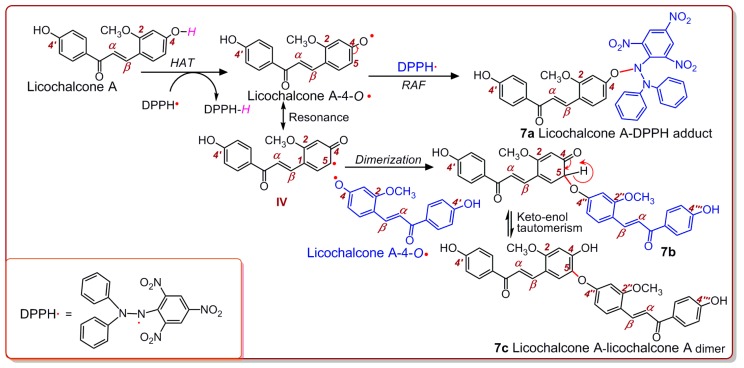
Proposed RAF reactions of echinatin when mixed with the DPPH• radical.

**Figure 8 molecules-24-00003-f008:**
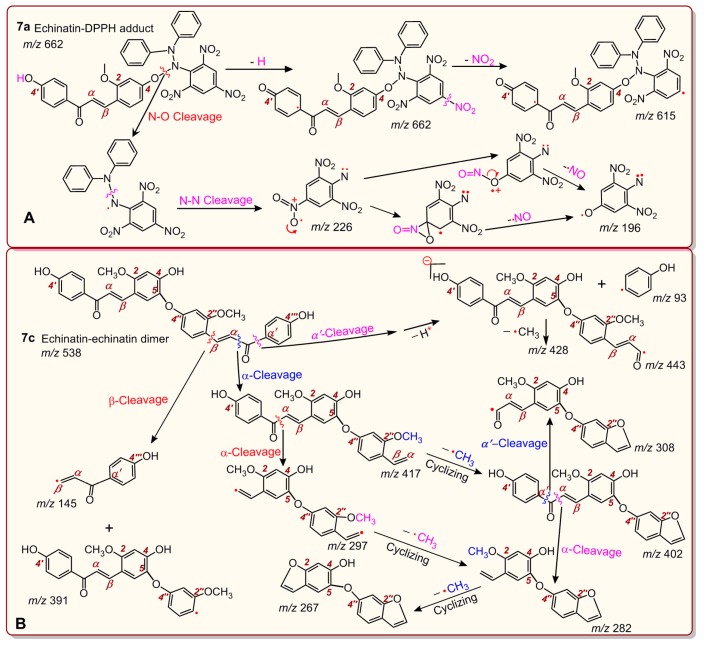
Full MS elucidation of the echinatin–DPPH adduct (**A**) and the echinatin–echinatin dimer (**B**). The MS spectra were obtained in the negative ion mode and the charge imposed by the MS field was not marked. Other reasonable cleavages should not be excluded in the MS elucidation.

**Figure 9 molecules-24-00003-f009:**
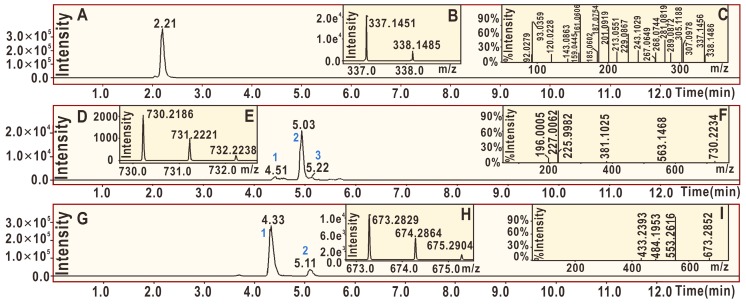
Main results of the UPLC-ESI-Q-TOF-MS/MS analysis of licochalcone A: (**A**) total ion chromatographic diagram of licochalcone A; (**B**) primary MS spectrum of licochalcone A; (**C**) MS/MS spectrum of licochalcone A; (**D**) total ion chromatographic diagram of the licochalcone A–DPPH adduct extracted by [C_39_H_33_N_5_O_10_-H]^−^; (**E**) primary MS spectrum of the licochalcone A–DPPH adduct; (**F**) MS/MS spectrum of the licochalcone A–DPPH adduct from peak 2 in (**D**); (**G**) total ion chromatographic diagram of the possible dimeric products of licochalcone A extracted by [C_42_H_42_O_8_-H]^−^; (**H**) primary MS spectrum of the licochalcone A–licochalcone A dimer; and (**I**) MS/MS spectrum of the licochalcone A–licochalcone A dimer from peak 1.

**Figure 10 molecules-24-00003-f010:**
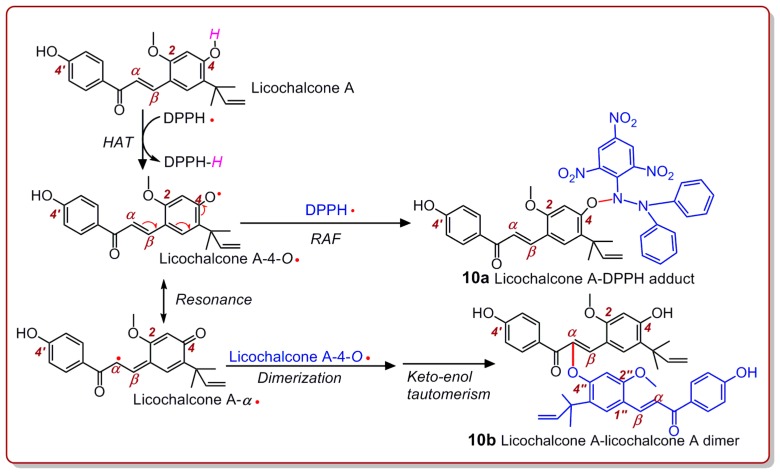
Proposed RAF reactions of licochalcone A when mixed with the DPPH• radical.

**Figure 11 molecules-24-00003-f011:**
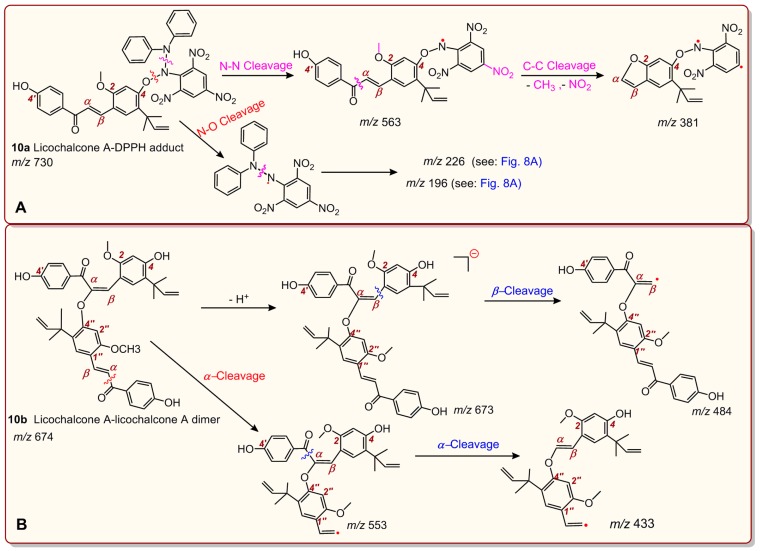
Full MS elucidation of the licochalcone A–DPPH adduct (**A**) and the licochalcone A–licochalcone A dimer (**B**). The MS spectra were obtained in the negative ion mode, and the charge imposed by the MS field was not marked. Other reasonable cleavages should not be excluded in the MS elucidation.

**Table 1 molecules-24-00003-t001:** IC_50_ values (μM) of echinatin and licochalcone A from the four sets of antioxidant colorimetric measurements.

Measurement	Echinatin	Licochalcone A	Trolox
Fe^3+^-reduction	338.0 ± 8.6 ^c^	133.2 ± 0.9 ^b^	87.5 ± 3.4 ^a^
Cu^2+^-reduction	228.1 ± 10.6 ^c^	129.1 ± 2.1 ^b^	67.5 ± 0.9 ^a^
PTIO•-scavenging	1276.5 ± 149.9 ^c^	617.4 ± 22.9 ^b^	270.9 ± 10.8 ^a^
DPPH•-scavenging	394.2 ± 67.5 ^c^	102.3 ± 3.6 ^b^	47.8 ± 2.6 ^a^

The IC_50_ value was defined as the final concentration of 50% radical inhibition or relative reduction power, calculated using linear regression analysis, and expressed as the mean ± standard deviation (SD) (*n* = 3). The linear regression was analyzed using the Origin Professional software (2017 version). IC_50_ values in the same row with different superscripts (^a^, ^b^, or ^c^) were significantly different (*p* < 0.05). Trolox was the positive control. The dose-response curves are shown in Appendix A.

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
