# Peer review of "Antioxidant Mechanisms of Echinatin and Licochalcone A"

_molecules, 2018, doi:10.3390/molecules24010003_

Reviewer 1 Report

Antioxidant Mechanisms of Echinatin and Licochalcone A

In this paper, authors report experimental results concerning the antioxidant properties of two important molecules that are found in the Chinese herbal medicine Gancao. Base don experimental evidence they proponed reaction mechanisms that are fine for me. I suggest two small modifications that could improve the manuscript.

First paragraph of Results and Discussion, authors said: “These antioxidant mechanisms are seemingly different from each other; however, all of these are essentially made up of two elemental reactions, i.e., an ET reaction and a PT reaction. The difference between them lies only in the sequence and collaboration” I do not understand this idea. Please complete or delete.

At the end of the manuscript authors said: “Of course, more work is required for identification in the future." They should be more specific. In my opinion answers were found and this investigation is quite complete  

Reviewer 2 Report

The paper “Antioxidant Mechanisms of Echinatin and Licochalcone A” reports about the antioxidant mechanisms of two natural components of Gancao. The aim of the paper was to assess the mechanisms responsible for the antioxidant activity of molecules under investigation. To this aim, colorimetric measurements were performed. Moreover, chromatographic techniques were used. The paper is well written: the introduction is appropriate and guide the reader to the paper, results and discussion are clearly presented and methods are sound. Hence, the paper is suitable for publication.

Author Response

Thank you very much.

Reviewer 3 Report

The authors successfully show the molecular mechanism how echinatin and licochlcone A transfer electron and proton using biochemical assay and mass analysis.

If the authors would provide more extensive conclusion with application regarding human health of these antioxidants chemical as active ingredients of foods and drugs, this manuscript would provide important information to readers.
